# Chitosan as an Alternative to Oil-Based Materials for the Fabrication of Lab-on-a-Chip

**DOI:** 10.3390/mi15030379

**Published:** 2024-03-12

**Authors:** Morgane Zimmer, Stéphane Trombotto, Emmanuelle Laurenceau, Anne-Laure Deman

**Affiliations:** 1Ecole Centrale de Lyon, INSA Lyon, CNRS, Université Claude Bernard Lyon 1, CPE Lyon, INL, UMR5270, 69130 Ecully, France; emmanuelle.laurenceau@ec-lyon.fr (E.L.); anne-laure.deman-him@univ-lyon1.fr (A.-L.D.); 2Ingénierie des Matériaux Polymères (UMR5223), Université Claude Bernard Lyon 1, CNRS, INSA Lyon, Université Jean Monnet, 69622 Villeurbanne, France; stephane.trombotto@univ-lyon1.fr

**Keywords:** chitosan, lab-on-a-chip, microfluidic, biomaterials, micro-drilling, hot embossing

## Abstract

Given the growing importance of lab-on-a-chip in a number of fields, such as medical diagnosis or environmental analysis, the fact that the current fabrication process relies mainly on oil-based polymers raises an ecological concern. As an eco-responsible alternative, we presented, in this article, a manufacturing process for microfluidic devices from chitosan, a bio-sourced, biodegradable, and biocompatible polysaccharide. From chitosan powder, we produced thick and rigid films. To prevent their dissolution and reduce their swelling when in contact with aqueous solutions, we investigated a film neutralization step and characterized the mechanical and physical properties of the resulting films. On these neutralized chitosan films, we compared two micropatterning methods, i.e., hot embossing and mechanical micro-drilling, based on the resolution of microchannels from 100 µm to 1000 µm wide. Then, chitosan films with micro-drilled channels were bonded using a biocompatible dry photoresist on a glass slide or another neutralized chitosan film. Thanks to this protocol, the first functional chitosan microfluidic devices were prepared. While some steps of the fabrication process remain to be improved, these preliminary results pave the way toward a sustainable fabrication of lab-on-a-chip.

## 1. Introduction

Microfluidics has promoted the precise manipulation of extremely small volumes of fluids and micro-objects, such as cells, in a controlled environment [1]. Thus, the miniaturization of conventional laboratory functions has enabled the development of lab-on-a-chip (LoC), opening up new perspectives in the fields of drug development [2,3], food and environmental analysis [4,5], medical diagnosis [6,7], and single-cell studies [8]. At first, LoC were based largely on glass, quartz, or silicon due to their well-known microfabrication methods derived from microelectronics [9,10,11]. Then, polydimethylsiloxane (PDMS) and associated soft lithography facilitated the fabrication process and prototyping [12,13]. Thanks to its high microfabrication resolution, optical transparency, and biocompatibility, PDMS now supports a broad range of chemical and biological applications [14,15]. However, PDMS has displayed a problematic absorption of small hydrophobic molecules [16], as well as the leaching of uncured polymer chains [17]. Transparent and biocompatible thermoplastics such as poly(methyl methacrylate) (PMMA) or cyclic olefin copolymer (COC) have also been used to fabricate LoC [14,18]. As opposed to PDMS [19], the approaches used for microfluidic fabrication include more complex methods that are more suitable for mass production [20]. For instance, hot embossing, injection molding, and sheet operations enable a high-volume production of microfluidic devices [21]. Due to the growing demand for fast and individualized point-of-care testing, the medical miniaturized diagnostic market is expected to represent USD 75.5 billion by 2027 [22], with already more than 1 billion tests used per year to diagnose COVID-19 [23], approximately 412 million for malaria [24], and 3 million for tuberculosis [25]. However, the increase in single-use tests also has a detrimental impact on the environment. The growing use of LoCs, made from oil-based materials, will generate environmental costs linked to both their manufacture and destruction. Few polymers can be efficiently recycled. Therefore, the plastic waste is either incinerated or landfilled, which can lead to air and water pollution, respectively [26].

In the wake of the growing environmental awareness, bio-based polymers have arisen as promising candidates for the fabrication of LoC. In particular, some possess an inherent biocompatibility, which is appealing for cell experiments, such as cell culture or sorting. Despite their potential, few papers have been published on the subject. The most widespread biopolymers, cellulosic materials, are successfully commercialized as microfluidic, paper-based devices. Fluids are transported through the device by capillary actions, and the resulting LoCs address a wide range of applications [27,28]. However, the transportation mode restricts the potential applications to biomarker detection or DNA extraction and does not allow for cell manipulation [29]. LoCs obtained from other biopolymers have been reported in the literature. Zein, a byproduct of ethanol production from corn, was processed into an LoC by Luecha et al. [30]. Although insoluble in aqueous solutions, zein films opacify when in contact with water, preventing monitoring by optical or fluorescent microscopy [31]. Yajima et al. used alginate hydrogels to produce microfluidic systems [32]. The resulting hydrogels are transparent, but their crosslinking ions leached unavoidably, resulting in a low cytotoxicity [33]. Silk hydrogels were developed by Zhao et al. [34]. Besides the complex but biocompatible fabrication process, silk hydrogels allow the diffusion of molecules. While it enables the perfusion of cells embedded in the hydrogel, it can also be an obstacle to other microfluidic functions. Ongaro et al. [35] reported the fabrication of an LoC made from poly(lactic acid) (PLA) extracted from the fermentation of corn or sugarcane starch. Although it exhibits a lower absorption of molecules than PDMS and a similar transparency [36], some authors consider PLA non-degradable under ambient conditions or when immersed in water [37,38]. In conclusion, at present, none of these biomaterials meet the specifications in terms of transparency, micropatterning, impermeability, biocompatibility, and biodegradability.

Chitosan is a non-toxic, biocompatible [39,40,41], biodegradable [42,43], and antimicrobial [44] polysaccharide composed of β-1,4-linked d-glucosamine and N-acetyl d-glucosamine units [45]. Obtained from the deacetylation of chitin, the second most widespread natural polymer on Earth, it is produced industrially by valorizing wastes from the seafood industry, already reaching several million tons per year [46]. The alluring properties of chitosan, including its ability to form films, have already been capitalized upon in a wide range of applications ranging from selective coatings for biosensors [47] and scaffolds for cell regeneration [39,48] to food packaging [49] and controlled drug release [50]. Chitosan films have also been micropatterned by soft lithography [39,51,52], e-beam lithography [40,53], oxygen plasma etching [54,55], or by electrodeposition [56,57], obtaining patterns in the range of tens of micrometers. Despite these results, as far as we know, chitosan has never been used to fabricate an entire microfluidic system. Indeed, on the one hand, the films described in the literature are only a few µm thick, which is incompatible with the fabrication of microchannels. On the other hand, the considerable swelling of chitosan film in water is an obstacle for microfluidics. To address this problem and prevent film dissolution, two strategies have been studied, as follows: crosslinking chitosan chains [58] or neutralizing them [59]. Most chitosan crosslinkers have a detrimental impact on the environment and the biocompatibility of the films, especially the most widely used, glutaraldehyde [60].

Here, we report a fabrication process of LoCs from chitosan. Our aim is to provide a more environmentally friendly alternative based on a bio-sourced and biodegradable material that could be used in applications requiring biocompatibility. To reduce the environmental impact and preserve the biocompatibility of chitosan, we have chosen to neutralize chitosan films. Different neutralization protocols have been tested to decrease the swelling of chitosan over time. The mechanical and physicochemical properties of the neutralized chitosan films have been studied, in particular, to estimate their Young’s modulus and transparency. These neutralized films were micropatterned by two distinct methods, micro-drilling and hot embossing. Each method was evaluated according to the resolution of the microfluidic channels obtained. LoCs were finalized by bonding the patterned chitosan films with photoresist to a glass slide or another chitosan film.

## 2. Materials and Methods

### 2.1. Materials

Chitosan powder, produced from shrimp shells, was purchased from Mahtani Chitosan Pvt. Ltd. (Veraval, India) from the batch number 20140503 of 244LG. Its average molecular weight (Mw) was estimated using size-exclusion chromatography (SEC) analysis to be 192 kg.mol^−1^, with a dispersity of 1.85. (Appendix A). Its degree of N-acetylation (DA) is 0.8%, according to 1H nuclear magnetic resonance (NMR) analysis. (Appendix A). Its water content (8.4% *w*/*w*) and its ash content (0.6% *w*/*w*) were determined using thermogravimetric analysis. (Appendix A).

Analytical grade acetic acid, sodium hydroxide, and ethanol were purchased from Sigma-Aldrich (Saint-Louis, MO, USA). The Ordyl SY 310 10 µm thick photoresist film was bought from MicroChemicals, GmbH (Ulm, Germany).

Deionized water (DI water) was obtained from a Purelab Chorus from Elga Veolia (High Wycombe, UK), with a conductivity of 18.2 Ω.

### 2.2. Chitosan Film Preparation

Chitosan was dissolved at 4% (*w*/*v*) in 2.4% (*v*/*v*) acetic acid solution at room temperature for 4 h. The chitosan solution was centrifuged at 4400 rpm for 6 min to remove bubbles. The solution was then cast into a 60 mm Petri dish and left in an oven overnight at 50 °C to evaporate the solvent. Then, the Petri dish was refilled with the chitosan solution and left in the oven at 50 °C for 24 h.

The dried chitosan films were neutralized by immersion in a sodium hydroxide solution. Different concentrations of NaOH and immersion times have been compared to determine the most efficient neutralization method. The films were then rinsed at least three times with DI water until the pH of the rinsing water reached seven. Lastly, the neutralized chitosan films were dried in the oven at 50 °C.

### 2.3. Chitosan Film Characterization

The thickness of the chitosan films after neutralization was measured with a digital caliper (Mitutoyo, Roissy, France). Swelling measurements were carried out as follows: First, dried chitosan films were cut into 2 × 3 cm^2^ pieces. After neutralization, their initial weights (W_0_) were measured. The films were then immersed in DI water and their weight (W_t_) was monitored as a function of time. The swelling was then calculated from the following equation:S_t_ (%) = (W_t_ − W_0_)/W_0_.(1)

The Fourier-transform infrared (FT-IR) spectra were obtained from Nicolet 6700 IR Spectrometer (Thermo Scientific, Waltham, MA, USA), equipped with a diamond accessory for attenuated total reflectance (ATR). A total of 64 scans were collected in the range of 4000–500 cm^−^^1^, with a resolution of 2 cm^−^^1^.

The water contact angle of neutralized films was estimated from five measurements of 0.8 µL DI water droplets using a PL-B771U camera (PixeLINK, Ottawa, ON, Canada) with an AF Micro Nikkor 60 mm objective (Nikon, Tokyo, Japan) and the Windrop++ software (1.02.01.01 GB).

To quantify the hardness and Young’s modulus of neutralized and non-neutralized chitosan films, fifteen nano-indentation measurements were performed on each sample using a G200 nano-indenter (Agilent Technologies, Santa Clara, CA, USA), with a Berkovich tip in diamond.

The transmittance of 0.3 mm thick chitosan films was measured on the ultraviolet–visible domain with a UVmc Spectrophotometer (Safas, Monaco, Monaco). The transmittance spectrum is given in the Appendix A. The corresponding transparency was then calculated by averaging the transmittance between 400 and 800 nm.

### 2.4. Channel Micropatterning

Two micropatterning methods, micro-drilling and hot embossing, were investigated to obtain microchannels in neutralized chitosan films.

For hot embossing, a brass master was first patterned by micro-drilling (Minitech Machinery Corp., Norcross, GA, USA) with the negative pattern of microchannels. The dimensions of the patterns were characterized using an optical microscope (Olympus BX51M associated with an Olympus SC50 camera, Evident Corporation, Tokyo, Japan) and a profilometer (Dektak 150 Stylus profiler, Veeco Instruments Inc., Tucson, AZ, USA). Channel molds with widths ranging from 123 µm to 1016 µm with an average height of 88 µm were used. Their dimensions are reported in Table 1 in Section 2.4. The master was then pressed onto neutralized chitosan films at a temperature of 65 °C and a force of 5.7 kN using a hydraulic press (JAS 105 BM 6181 RONDOL Hi-Force Hydraulic tools, Northants, UK). After 1 h, the system was cooled to a temperature of 40 °C before demolding the newly imprinted chitosan films from the molds.

For micro-drilling, the neutralized chitosan films were engraved directly using a 100, 200, or 300 µm diameter micro-drill piloted by a Desktop 3-Axis micro-milling machine (Minitech Machinery Corp., Norcross, GA, USA), controlled by a Nakanishi E3000 controller (Nakanishi Inc., Tochigi, Japan).

The width resolution of the microchannels, obtained using the two microfabrication methods, was determined by measuring the channel width five times along a channel with an optical microscope (Olympus BX51M, Evident Corporation, Tokyo, Japan) equipped with a camera (Olympus SC50, Evident Corporation, Tokyo, Japan). Their depth resolution was measured with a profilometer (Dektak 150 Stylus profiler, Veeco Instruments Inc., Tucson, AZ, USA) with a tip radius of 12.5 µm. As each channel dimension was replicated three times, ANOVA analyses were performed between measurements carried out on similar channels to determine if the micropatterning methods were reproducible. The *p*-values are reported in the Results section.

The rugosity at the bottom of the micro-drilled or hot embossed channels was estimated from the root mean square deviation measured on three areas of 30 × 30 µm^2^ mapped using Atomic Force Microscopy (AFM MFP-3D Asylum Research, Oxford Instrument, Abingdon, UK), equipped with a 40 N.m^−1^ Arrow NCR probe (Non-contact/Tapping mode–Reflex coating).

### 2.5. Microfluidic Device Preparation

After the micropatterning of neutralized chitosan films, the microfluidic systems were sealed by bonding the micropatterned films to another substrate. The solid Ordyl SY 310 photoresist (MicroChemicals GmbH, Ulm, Germany) was laminated on the micropatterned film at 50 °C. Next, the film covered in photoresist was pressed to either a glass slide or another chitosan film using the hydraulic press at 65 °C and 0.5 kN for 10 min (hydraulic press, JAS 105 BM 6181 RONDOL Hi-Force Hydraulic tools, Northants, UK). Once the system is cooled down to below 40 °C, the photoresist is crosslinked by UV light at 365 ± 5 nm in a UV-KUB 2 (KLOE, Saint-Mathieu-de-Tréviers, France) for 1 min at 35 mW.cm^−2^. At each step, the deposition of the photoresist was monitored under an optical microscope.

Liquids were injected with a PHD Ultra syringe pump (Harvard Apparatus, Holliston, MA, USA) attached to the chitosan microfluidic device with connectors from the microfluidic ChipShop (Jena, Germany). An AD409 digital microscope (Andonstar, Shenzhen, China) was used to monitor the microchannels under injection.

## 3. Results and Discussion

### 3.1. Chitosan Film Preparation and Characterization

Chitosan films were successfully obtained using solvent casting. The polysaccharide is water soluble solely in acidic conditions, as the amino groups of its d-glucosamine units are protonated to –NH_3_^+^ (Figure 1a). Chitosan films of 470 µm thickness were obtained after the evaporation of the solvent, with 60 µm standard deviation between samples.

As the amino groups of chitosan were still protonated, chitosan films were soluble in aqueous solutions. To reverse their protonation state to –NH_2_, chitosan films were entirely immersed in sodium hydroxide solution (Figure 1a). Afterwards, the films were thoroughly rinsed to remove any water-soluble molecules and bring the pH back to neutral. The neutralization of chitosan films was tested for two molar concentrations of sodium hydroxide (0.1 M or 1 M) and two immersion times (10 min or 45 min).

After neutralization, the chitosan film thickness was assessed to be, on average, 400 µm, with a 50 µm standard deviation between samples. Each chitosan film also exhibited a variation in the thickness along the surface of up to 30 µm. The thickness of chitosan decreased after the neutralization, as acetic acid molecules between chitosan chains were also neutralized and removed by rinsing [61]. Therefore, the space between polymer chains was reduced, which decreased the overall thickness of the film [59].

This fact was corroborated by the IR spectra of the chitosan films both before and after neutralization with 1 M NaOH for 45 min, as shown in Figure 1b. The change in protonation state was marked by a shift of the peak of amide II at 1550 cm^−1^ to a higher wavenumber, characteristic of the neutralization process of chitosan. The disappearance of the COO- peak at 1640 cm^−1^ was the result of the removal of the acetic acid. All of these phenomena impacted the intermolecular interactions, which were observed by the shift of the N-H and O-H vibrations band around 3200 cm^−1^ to higher wavenumbers [62,63].

We investigated the swelling of neutralized and non-neutralized films as a function of time. The results are reported in Figure 2. As expected, without neutralization, chitosan films exhibited the highest swelling rate until they lost their cohesion and partially dissolved after 30 min. Depending on the NaOH concentration and their immersion time, chitosan films displayed different behaviors once submerged in deionized water. For all neutralized films, the water absorption was faster in the first 10 min before slowing down. Furthermore, the films immersed in 1 M sodium hydroxide reached a plateau of swelling after 30 min. All in all, the swelling rate decreased for a higher concentration of sodium hydroxide and for a longer immersion time in sodium hydroxide. These results are consistent with the work of Chang et al. [59]. It is also interesting to note that the variation between samples was lower for a higher concentration and a longer immersion in sodium hydroxide, indicating better reproducibility.

The best neutralization conditions, in terms of the swelling rate (49 ± 1% after 30 min in deionized water) of chitosan films and its reproducibility, were 45 min immersion in 1 M NaOH. In the following characterization studies, neutralized chitosan films will refer to neutralization under these conditions. However, this step should be optimized to avoid the absorption of small molecules into the chitosan film due to its swelling. This may be problematic for microfluidic applications that require the precise control of small molecule concentrations (such as drugs in chemosensitivity studies or pollutants in environmental studies). As observed for PDMS [16,64], due to its permeability, it tends to adsorb small molecules and modifies the drug concentrations in the channels. In addition, it would be interesting to evaluate and characterize the potential release of small polymer chains or molecules from the neutralized chitosan film due to swelling in an aqueous medium.

### 3.2. Properties of Neutralized Chitosan Films

Optical transparency is essential for the microscopic monitoring of micro-objects, such as in cell experiments. Chitosan films appeared optically transparent with a light-yellow coloration. The transmittance of neutralized chitosan films was characterized between 400 nm and 800 nm, with an average value of 75%. As expected from the observable coloration, chitosan films more strongly absorbed wavelengths below 500 nm [65]. However, this is lower than the transmittance of PDMS and thermoplastics, which were evaluated to be more than 90% [14,18]. For bio-based polymers, to our knowledge, only PLA transparency was characterized for LoC applications, reaching 92% [36].

The surface roughness and hydrophobicity of neutralized chitosan films were measured using AFM and water contact angle, respectively. The surface roughness was about 72 ± 15 nm (Appendix A). For comparison, glass slides [66,67] or commercially available PMMA and COC wafers [68] displayed a surface roughness between 5 and 30 nm. The surface of neutralized chitosan films displayed a water contact angle of 99.5 ± 2.4°, indicating their hydrophobic character.

We characterized the mechanical properties of the films using nano-indentation. Non-neutralized chitosan films showed a Young’s modulus of 6.4 ± 0.4 GPa and a hardness of 0.25 ± 0.03 GPa, measured at 23.9 °C. After neutralization, chitosan films exhibited a Young’s modulus of 7.3 ± 0.2 GPa and a hardness of 0.4 ± 0.03 GPa. The increment in strength after neutralization may be explained by the removal of the water and acetic acid molecules remaining in the chitosan films, as well as the resulting increases in interactions between chitosan chains [59,69]. Indeed, these small molecules could have acted as plasticizers and weakened the polymer chain interactions. Neutralized chitosan films demonstrated mechanical properties closer to thermoplastics like PMMA [70] with a Young’s modulus of 5 GPa and COC [71] with 3 GPa than PDMS [72], which has a Young’s modulus between 1.61 and 2.01 MPa. These results, therefore, guided the selection of the film micropatterning method.

### 3.3. Micropatterning of Chitosan Films

Two different micropatterning methods were investigated to produce microfluidic channels on neutralized chitosan films, hot embossing, a standard technique for patterning thermoplastics, and micro-drilling, recently deployed for thermoplastics. These patterning methods were compared based on the width and depth of the obtained channels and their reproducibility.

For hot embossing, masters were fabricated by micro-drilling negative patterns of microchannels on brass disks with a 500 µm diameter micro-drill. By mounting the micro-drill on a spindle and rotating it at high speed, one can carve through rigid materials such as thermoplastics and brass. Micro-drilling offers a faster and less complex fabrication process than UV-LIGA for masters [73]. The master’s channels ranged in width from 123 to 1016 µm, with a length of 1 cm and a height of 88 µm. Their dimensions are given in Table 1. A microscopic image of a brass master with a 521 µm width is shown in Figure 3a.

The masters were pressed on the chitosan films (Figure 3b) according to the parameters specified in Section 2.4. The machining marks from the mold were transferred to the bottom of the channel, as shown in Figure 3c for the 521 µm wide channel. The surface roughness was estimated using AFM, at the bottom of a 1000 µm wide channel, to be 142 nm.

The heights and widths of the imprinted channels were characterized using a profilometer and an optical microscope, respectively. The channels flared out between the bottom and the top of the channel. The images and profile of a channel obtained from the 521 µm wide master are shown in Figure 3d and Figure 3c, respectively. The channel width varied from 505 µm at its bottom to 868 µm at the top. Thus, the widths of the different channels at the bottom and the top of the channels were measured and reported in Table 1. One can notice that, at the bottom, the channels were imprinted with accuracies ranging from 90% for the narrowest channel (123 ± 6 µm wide) to 99.6% for the widest channel (1016 ± 2 µm). The low standard deviations obtained at the bottom of the channels attested to the regularity of the patterning along the 1 cm long channels.

However, the channels obtained from hot embossing did not have a rectangular section, as they widened upwards. For instance, for a mold of 521 ± 3 µm, the width of a hot embossed channel varied between 506 ± 1 µm at the bottom and 883 ± 32 µm at the top. Moreover, there were huge variations of the width at the top of the channel along each channel and between similar channels and a variability of the accuracy of the depths obtained (between 99% and 33%).

From those results, chitosan did not adapt to the shape of the mold as freely as COC or PMMA. Therefore, it was not able to reproduce huge form factors as finely, deterring the use of hot embossing to micropattern chitosan films. Nevertheless, it is worth noticing that chitosan seemed to be able to imprint smaller form factors at the master’s defects.

As micro-milling is a powerful tool for engraving hard materials, we have used it to directly engrave channels in chitosan films. The resulting mechanical micro-drilling is limited in resolution by the diameter of the drill, with the smallest drill available measuring 10 µm in diameter, but offers great versatility since it is possible, for example, to easily vary the height of the channels in a single microsystem.

Several microchannels were produced by micro-drilling on neutralized chitosan films. (Figure 4a). Each sample was then cleaned with ethanol to remove any chitosan fragments left after the drilling. A drill of 200 µm in diameter was instructed to generate 250, 500, and 1000 µm wide channels, whereas 105 and 200 µm channels were engineered from a drill of 100 µm diameter. These channels were 200 µm deep, as illustrated by the profile in Figure 4b. The slopes of the channel walls and their asymmetry are artifacts due to the abrupt change in the height and the curvature radius of the profiling tip. In contrast, with hot embossed channels, micro-drilling created microchannels with sharp edges, as depicted in Figure 4b,c. According to the width measurements reported in Table 2, except for the 105 µm wide channels, which have an 87% accuracy with respect to the instruction, the other micro-drilled channels have an accuracy of over 97%. The close similarity between the instructions and the resulting widths is complemented by the regularity of these widths over the 1 cm length of every channel. As a point of fact, the micro-drilled channels presented an average of 2 µm standard deviation. The variations between channels with the same instruction could be due to slight variations in the diameter of the micro-drills.

The roughness of the micro-drilled surface (Figure 4d) depends on the drill dimensions along with the program parameters, such as rotating speed or distance of overlap. On a 1000 µm wide channel, the roughness was estimated at 375 nm for a 10,000 rpm and 0.02 mm overlapping. This could be improved, as the literature reported a roughness of between 100 and 200 nm for optimized micro-drilling on PMMA or COC [68]. However, one has to be careful about the parameters because micro-drilling creates heat that can partially deform the substrates [74].

To analyze the depth resolution for micro-drilling, micro-channels of 350 µm in width, with a depth varying from 50 µm to 200 µm, were produced with a 300 µm diameter drill. To disregard the variation of film thickness in the depth measurements, all measurements were performed on the 0 µm height-calibrated area on the chitosan surface. Results are reported in Table 2. The depths obtained display differences against the instructions going from 5 µm for the 50 µm channel to 25 µm for the 200 µm one. As the channels are engraved with multiple drill passes, a constant error at each pass could accumulate, resulting in an average accuracy of 91% for all depths. Except in one case, all channels were less deep than expected, which could have been planned for before the fabrication process.

We produced microchannels of various dimensions using two distinct fabrication methods. In light of the results obtained with hot embossing and micro-drilling for engraving microchannels in neutralized chitosan films, micro-drilling offered a superior performance. For similar accuracy and regularity of width, the channel edges were better defined for micro-drilling, as opposed to the deformed ones from hot embossing. Furthermore, micro-drilling allows for more flexible designs, in terms of heights and faster prototyping. Thus, the micro-drilling approach was favored to produce microfluidic prototypes.

### 3.4. Microfluidic Protypes

On neutralized chitosan films, microchannels and holes for tubing connection have been micro-engraved. Microfluidic systems were obtained by bonding the chitosan films to another substrate according to the steps described in Figure 5a. The successive bonding steps were carefully monitored using optical microscopy, as illustrated in Figure 5b. As observed in Figure 5(b.ii), the laminated photoresist has been removed above the micropatterned patterns, except a small portion close to the border. After the pressing step (Figure 5(a.iii,b.iii)), the photoresist slightly overflowed into the channel, probably due to the pressure and temperature. This should not hinder the microfluidic functions, as the thickness of the photoresist (10 µm) is much lower than the height of the channel. However, the lamination and pressing steps should be optimized to avoid overflow of the photoresist in the microchannel. Finally, chitosan films were bonded to either a glass slide or another neutralized chitosan film (Figure 5(a.iii)) to seal the microfluidic devices.

Tubes were connected to the outlet and inlets of the devices using connectors, and dyed aqueous solutions were injected into the chitosan/glass (Figure 5(c.i)) and chitosan/chitosan (Figure 5(c.ii)) microfluidic systems. Other geometries were fabricated as a serpentine-like channel, used to promote the efficient mixing of different components, while reducing the overall length of the system. As shown in Figure 6, a co-flow of blue and red dyed aqueous solutions, observed at the beginning of the 500 µm wide channels, tends to mix along the serpentine.

We tested the resistance of photoresist bonding on microfluidic devices with straight channels, 500 µm wide and 150 µm high. First, we assessed the bond strength as a function of the flow rate applied to chitosan/glass systems. The flow rate of an aqueous solution was increased incrementally until 400 mL.h^−1^. Under these conditions, the photoresist bonding was still functional on three different devices. We also tested the effectiveness of photoresist bonding by subjecting chitosan/glass and all-chitosan devices to slow flow rates over long periods. An aqueous solution was injected into the microfluidic channels at a flow rate of 100 µL.h^−1^ for several hours. After 24 h of flow, the microfluidic systems had not leaked and the flow continued to circulate in the channels. (Appendix A). Consequently, these results are promising for microfluidic applications requiring a high flow rate or a long running time.

## 4. Conclusions

In this work, we present our first results in developing a fabrication process for LoCs that is more environmentally friendly than the traditional processes based on PDMS or thermoplastics. We have chosen chitosan, a bio-sourced, biocompatible, and biodegradable polymer, as the base material for the microfluidic systems. Through a film casting process, we produced thick and rigid chitosan films. After a neutralization step to prevent their dissolution in water, chitosan films exhibited 75% transparency and a Young’s modulus of 6.4 ± 0.4 GPa for 400 ± 50 µm of thickness. Due to the neutralization step, their swelling was decreased to 49 ± 1% after 30 min in deionized water. Then, the films were micropatterned using two methods, namely hot embossing and micro-drilling, which enabled us to compare their respective resolutions in channel height and width. Whereas hot embossing was not able to faithfully imprint the channels on chitosan films, micro-drilling offered a better accuracy and reproducibility for micropatterning, even if the resolution is limited by the drill diameter. The microfluidic channels were then sealed by bonding the micropatterned chitosan to a glass slide or another neutralized chitosan film using a biocompatible photoresist. We evaluated the bonding stability of the devices for flow rates of up to 400 mL.h^−1^ and for low flow rates perfused for several hours. With micro-drilling, functional microfluidic devices with various designs were prepared. To the best of our knowledge, they were the first microfluidic devices made from chitosan. Although there are still some steps of the fabrication process to be improved in order to further reduce the swelling and enhance the device bonding, these preliminary results open new opportunities for a sustainable fabrication of LoCs. Thanks to the biocompatibility of chitosan, among its other properties, more complex functions can be studied, including cell manipulation and cell culture.

## Figures and Tables

**Figure 1 micromachines-15-00379-f001:**
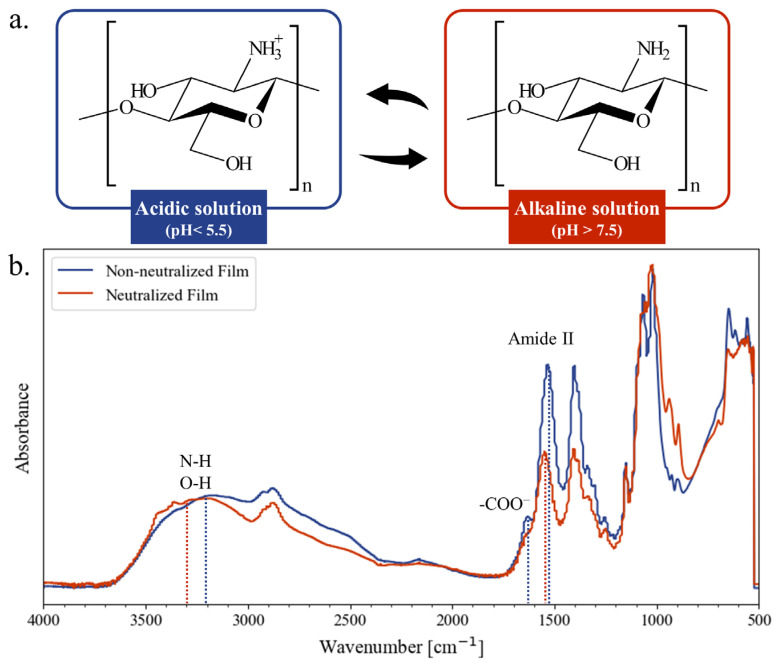
(**a**) Protonation states of the chitosan macromolecules; (**b**) normalized FT-IR spectra of the absorbance of chitosan films before and after neutralization at 1 M NaOH for 45 min.

**Figure 2 micromachines-15-00379-f002:**
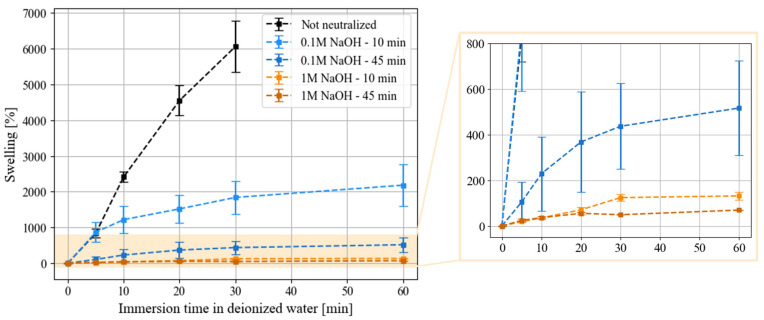
Swelling measurements as a function of the immersion time of chitosan films in deionized water. The neutralization applied is defined by the molar concentration of sodium hydroxide (x NaOH) and the immersion time in alkaline solution (y min). Each measurement was performed in triplicate.

**Figure 3 micromachines-15-00379-f003:**
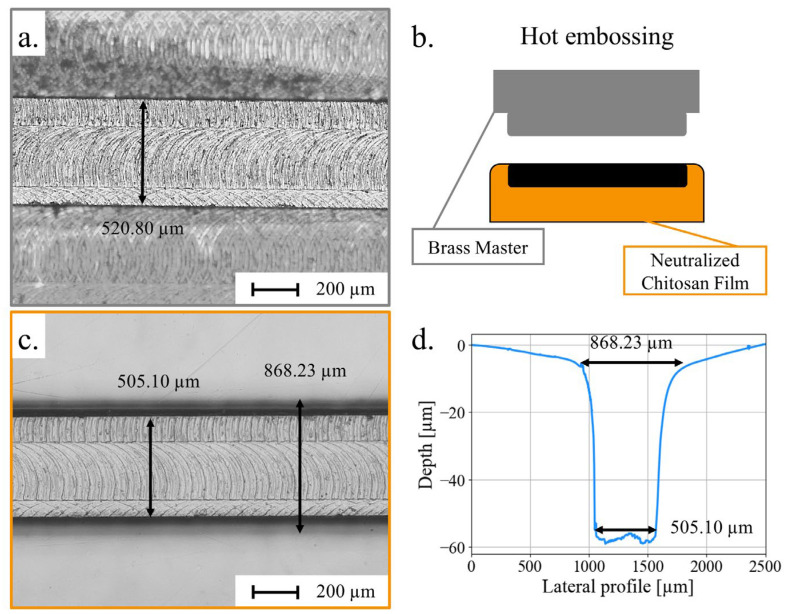
(**a**) Microscopic image of a brass master; (**b**) schematic of hot embossing protocol to produce microchannels by pressing a brass master onto a neutralized chitosan film; (**c**) microscopic image; and (**d**) depth profile of the corresponding imprinted microchannel in chitosan film.

**Figure 4 micromachines-15-00379-f004:**
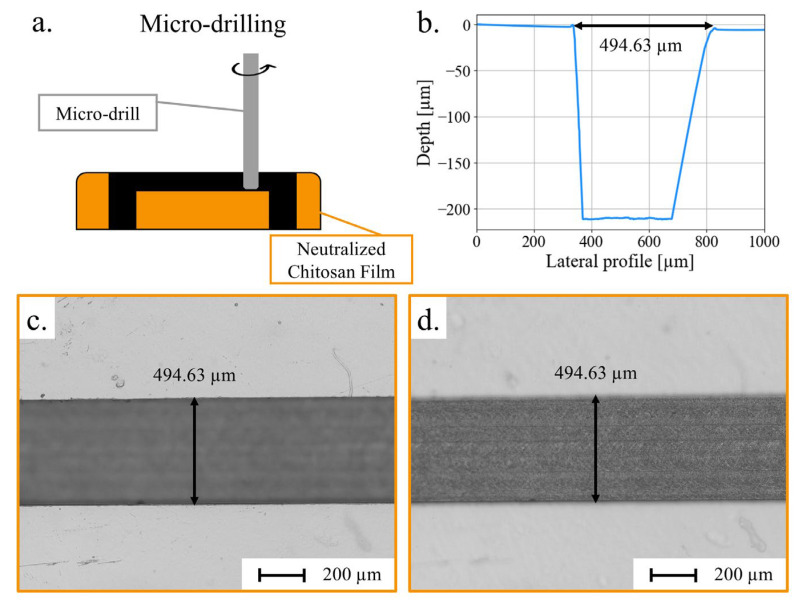
(**a**) Schematic of micro-drilling protocol to produce microchannels on a neutralized chitosan film; (**b**) depth profile; (**c**) microscopic image of a micro-drilled channel in chitosan film; and (**d**) microscopic image of the bottom of the micro-drilled channel.

**Figure 5 micromachines-15-00379-f005:**
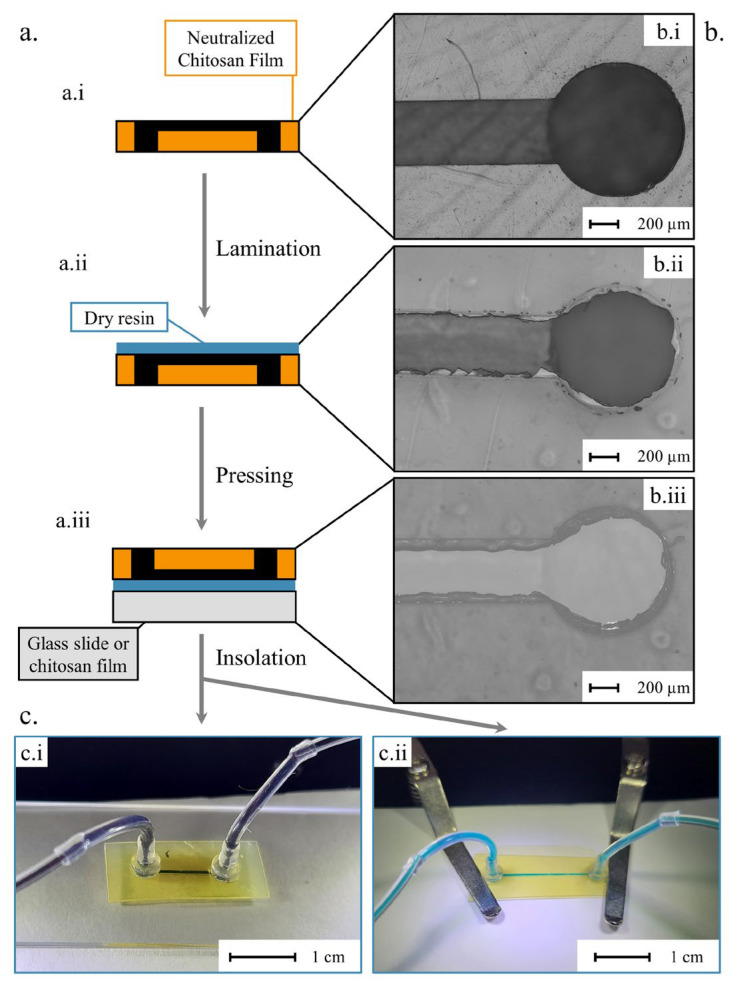
(**a**) Schematics of the bonding protocol, (**a.i**) the neutralized and micropatterned film to be bond, (**a.ii**) the deposition of the photoresist on the film and (**a.iii**) the bonding to a glass or chitosan substrate. (**b**) Microscopic images of a micropatterned chitosan film at different steps, (**b.i**), (**b.ii**) and (**b.iii**), corresponding respectively to (**a.i**), (**a.ii**) and (**a.iii**). (**c**) Photos of a chitosan/glass (**c.i**) and all-chitosan microfluidic systems (**c.ii**) with slide clips to maintain the system under the microscope, with channels of 350 µm in width and 150 µm in height, injected at 1 mL.h^−1^.

**Figure 6 micromachines-15-00379-f006:**
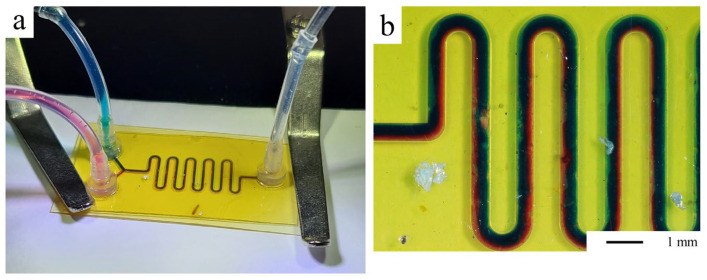
(**a**) Microfluidic system with red and blue aqueous solutions injected at 1 mL.h^−1^ in 500 µm wide and 150 µm deep channels in a full chitosan chip with slide clips to maintain the system under the microscope. (**b**) Microscopic image of the channel.

**Table 1 micromachines-15-00379-t001:** Hot embossing characterization: width measurements using optical microscopy and height measurements using profilometer, both averaged over the length of the channels. For each of the five sizes, the master and three hot embossed channels at 65 °C and 5.7 kN for 1 h were examined. The masters were designed with channels with widths between 105 and 1000 µm and a height of 100 µm. ANOVA tests were performed for each set of hot embossed channels. If no value is given, the *p*-value was negligible.

Master Width [µm]	123 ± 6	219 ± 4	270 ± 3	521 ± 3	1016 ± 2
Sample width at the bottom of the channel [µm]	120 ± 5	211 ± 5	262 ± 3	508 ± 4	1009 ± 13
	122 ± 4	210 ± 5	273 ±10	506 ± 1	1013 ± 12
	110 ± 7	215 ± 5	262 ± 5	512 ± 6	1009 ± 5
	*p* = 0.014	*p* = 0.29	*p* = 0.044	*p* = 0.13	*p* = 0.73
Sample width at the top of the channel [µm]	183 ± 30	609 ± 92	611 ± 208	883 ± 156	1240 ± 33
	443 ± 31	457 ± 62	602 ± 97	883 ± 32	1230 ± 54
	233 ± 51	391 ± 34	473 ± 30	712 ± 20	1278 ± 48
Master height [µm]	88 ± 1	88 ± 2	88 ± 1	88 ± 1	88.5 ± 0.1
Sample depth [µm]	83 ± 8	79 ± 5	41 ± 42	83 ± 8	87 ± 2
	89 ± 4	44 ± 9	77 ± 5	61 ± 1	54 ± 9
	29 ± 5	53 ± 5	68 ± 21	68 ± 21	67 ± 14

**Table 2 micromachines-15-00379-t002:** Micro-drilling characterization: width measurements using optical microscopy averaged over the length of the channel (five measurements per channel) and depth measurements using a profilometer at the calibration point. For each test, the channels were made in triplicate. ANOVA tests were performed for each set of micro-drilled channels. If no value is given, the *p*-value was negligible.

Width Instruction [µm]	105	200	250	500	1000
Measured width [µm]	114 ± 1	203 ± 2	245 ± 2	499 ± 2	988 ± 7
	119 ± 2	206 ± 2	245 ± 2	500 ± 5	1003 ± 4
	117 ± 2	205.5 ± 0.8	249 ± 2	497 ± 4	985 ± 13
	*p* = 0.0056	*p* = 0.028	*p* = 0.0055	*p* = 0.54	*p* = 0.018
Depth instruction [µm]	50	100	150	200	
Measured depth [µm]	45.7 ± 0.3	105.8 ± 0.2	138.0 ± 1	174.6 ± 0.2	
	44.6 ± 0.4	96.7 ± 0.3	136.3 ± 0.5	175.5 ± 0.2	
	47.9 ± 0.4	92.9 ± 0.2	138.9 ± 0.6	173.3 ± 0.2	

## Data Availability

Data will be made available on request.

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
