# Peer review of "Chitosan as an Alternative to Oil-Based Materials for the Fabrication of Lab-on-a-Chip"

_micromachines, 2024, doi:10.3390/mi15030379_

Round 1

Reviewer 1 Report

Comments and Suggestions for Authors

This article presents a fabrication process for microfluidic devices from chitosan as an eco-responsible alternative to the oil-based polymers traditionally used. The focus was given to the investigated of the film neutralization step and characterized the mechanical and physical properties of the resulting films. This article developed a fabrication process for lab on a chip that is more environmentally friendly than traditional processes. However, the paper needs very significant improvement. There are some concerns for authors to address and correct.

1.     The abstract should summarize the entire article. The abstract of this article describes only the procedure and results of the experiment. Background, significance and outlook should be added.

2.     When presenting the background of the research, more references to recent publications should be given.

3.     Microscopic images of the chip after a flow rate of 400 mL/h or 24 hours of flow should have been included in the supporting information.

4.     Authors should verify the correct use of units in their articles. Some units are missing spaces between them and numbers, or there are problems with the formatting of some units.

5.     Authors are encouraged to add specific applications such as material synthesis or cell culture to complete the data.

6.     Some supporting information requires a description in the body of the text. Such as Figure S1 (c), Figure S2, etc.

Author Response

You will find in the attached file the detailed response to the remarks made by reviewer 1 and the corresponding changes made to the manuscript. 

Reviewer 2 Report

Comments and Suggestions for Authors

The authors described a fabrication process for microfluidic devices from chitosan, a bio-sourced, biodegradable, and biocompatible polysaccharide, as an eco-responsible alternative to oil-based polymers. The authors also investigated a film neutralization step and characterized the mechanical and physical properties of the resulting films from 100 μm to 1000 μm wide. Further, chitosan films with micro-drilled channels were bonded using a biocompatible dry photoresist on a glass slide or another neutralized film. However, several chitosan-based LoCs are used for numerous applications, what is the uniqueness of the proposed work? How is this process different from the existing methods? What is the sensitivity, reproducibility, and dynamic range that can be incorporated into this proposed LoC?  Below are a few of the comments that need to be addressed before the paper is accepted for publication.

1)     Page 1, Line 31, “microelectronics. [9-11]” should be “microelectronics [9-11].”

2)     Page 1, Line 38,” LoC. [13, 17] should be “LoC [13, 17].”

3)     Page 1, Line 42, $75.5B, ‘B’ should be elaborated.

4)     Page 2, Line 72, please add the relevant reference.

5)     The introduction section used to be more comprehensive and should be discussed rigorously, covering key aspects of the scope including, the significance of microfluidics, nanomaterials with oil-based synthesis, and different fabrication techniques.

6)     Please avoid repetitive and incomplete sentences in the introduction “fabrication process of LoC from chitosan”

7)     Figure 1(a) & (b) resolution should be improved and the fonts of captions & legends need to be increased.

8)     Section 3.3 on micropatterning of chitosan films should be elaborated.

9)     Authors have not discussed the challenges and limitations of the process.

10) An introduction section should be added with relevant references. The author may consider a few recent state-of-the-art:

10.1016/j.apmt.2018.03.005; 10.3390/mi12030319; 10.3390/bios12070543; doi.org/10.1016/j.bej.2023.109027; 10.3390/bios13030412; 10.1038/s41598-020-65995-x

doi.org/10.2144/btn-2022-0091; 10.1016/j.coelec.2021.100800; 10.1016/j.bej.2023.109027; doi.org/10.1016/j.sna.2023.114385

11) In section 4.3, please describe the significance of microstructure design in serpentine structure and its importance in microfluidic technology.

12) Please include applications and the future scope of the work.

13) There are lots of spelling mistakes in the manuscript. Please correct them.

14) Overall English language should be checked and improved rigorously.

Comments on the Quality of English Language

NA

Author Response

You will find in the attached file the detailed response to the remarks made by reviewer 2 and the corresponding changes made to the manuscript. 

Round 2

Reviewer 1 Report

Comments and Suggestions for Authors

The revised manuscript can be accepted.